# YES1 Kinase Mediates the Membrane Removal of Rescued F508del-CFTR in Airway Cells by Promoting MAPK Pathway Activation via SHC1

**DOI:** 10.3390/biom13060949

**Published:** 2023-06-06

**Authors:** Patrícia Barros, Ana M. Matos, Paulo Matos, Peter Jordan

**Affiliations:** 1Departamento de Genética Humana, Instituto Nacional de Saúde Doutor Ricardo Jorge, 1649-016 Lisboa, Portugal; patricia.barros@insa.min-saude.pt (P.B.); ana-matos@campus.ul.pt (A.M.M.); peter.jordan@insa.min-saude.pt (P.J.); 2BioISI—Biosystems & Integrative Sciences Institute, Faculty of Sciences, University of Lisboa, 1749-016 Lisboa, Portugal

**Keywords:** Cystic Fibrosis, F508del-CFTR, plasma membrane half-life, MAPK pathway

## Abstract

Recent developments in CFTR modulator drugs have had a significant transformational effect on the treatment of individuals with Cystic Fibrosis (CF) who carry the most frequent F508del-CFTR mutation in at least one allele. However, the clinical effects of these revolutionary drugs remain limited by their inability to fully restore the plasma membrane (PM) stability of the rescued mutant channels. Here, we shed new light on the molecular mechanisms behind the reduced half-life of rescued F508del-CFTR at the PM of airway cells. We describe that YES1 protein kinase is enriched in F508del-CFTR protein PM complexes, and that its interaction with rescued channels is mediated and dependent on the adaptor protein YAP1. Moreover, we show that interference with this complex, either by depletion of one of these components or inhibiting YES1 activity, is sufficient to significantly improve the abundance and stability of modulator-rescued F508del-CFTR at the surface of airway cells. In addition, we found that this effect was mediated by a decreased phosphorylation of the scaffold protein SHC1, a key regulator of MAPK pathway activity. In fact, we showed that depletion of SHC1 or inhibition of MAPK pathway signaling was sufficient to improve rescued F508del-CFTR surface levels, whereas an ectopic increase in pathway activation downstream of SHC1, through the use of a constitutively active H-RAS protein, abrogated the stabilizing effect of YES1 inhibition on rescued F508del-CFTR. Taken together, our findings not only provide new mechanistic insights into the regulation of modulator-rescued F508del-CFTR membrane stability, but also open exciting new avenues to be further explored in CF research and treatment.

## 1. Introduction

The Cystic Fibrosis transmembrane conductance regulator (CFTR) protein is a member of the ATP-binding cassette transporter superfamily that functions as a cAMP-activated chloride and bicarbonate ion channel at the apical membrane of epithelial cells [1]. Mutations in the CFTR gene lead to Cystic Fibrosis (CF), a lethal disease characterized by pancreatic insufficiency, increased salt concentration in sweat, male infertility, and progressive lung disease [2]. Among the many CFTR pathogenic mutations identified to date, the deletion of phenylalanine 508 (F508del) is by far the most prevalent, occurring in up to 85% of CF individuals [3]. F508del causes CFTR to misfold, leading to the retention of most of the mutant protein at the endoplasmic reticulum (ER) and its premature degradation by the ER quality control machinery (ERQC) associated with proteasomes [4]. This causes a drastic reduction in the number of mutant channels reaching the plasma membrane (PM). In addition, the mutant protein exhibits a considerable gating defect due to an abnormal persistence in the closed state [5]. These combined defects result in a residual level of chloride secretion and epithelial fluid transport in the airway, leading to the production of a thick mucus that promotes airway obstruction, chronic bacterial colonization and inflammation, severe lung disease, and ultimately, respiratory failure [6].

In recent decades, the recognition that F508del-CFTR molecular defects could be targeted by chemical compounds has led to considerable efforts to develop and approve effective drugs, now termed “CFTR modulators”, to treat CF disease [7]. The most successful thus far, Kaftrio (trade name in Europe; Trikafta in the U.S.), combines three modulators: two “correctors” to improve F508del-CFTR folding (tezacaftor and elexacaftor (also known as VX-661 and VX-445, respectively)), and another to improve F508del-CFTR gating (ivacaftor (also known as VX-770), termed “potentiator”) [8]. Kaftrio has been considered groundbreaking in the CF clinical practice, with significant clinical benefit in patients 12 years and older who have at least one allele with the F508del mutation [9].

However, Kaftrio is still unable to fully restore F508del-CFTR function [3]. Even in vitro, the combination of modulators only rescues F508del-CFTR to approximately 60% of wild-type (wt) CFTR function in non-CF cells [10,11]. A major issue with modulator-rescued F508del-CFTR (rF508del-CFTR) is the considerably reduced half-life of the protein at the cell surface [10,12,13,14]. Although corrector drugs enable a significant amount of mutant protein to evade the ERQC, the rescued channels fail to completely escape the peripheral protein quality control (PPQC) machinery. The PPQC removes defective proteins from the PM, promoting their lysosomal degradation [13]. In the PM, wt-CFTR is internalized slowly via clathrin-mediated endocytosis (CME), but most of the internalized protein is rapidly recycled back to the PM [12,13,15]. In contrast, rF508del-CFTR is internalized by CME at a much faster rate [15,16,17], which is facilitated by the presence of chaperones and co-chaperones, and most of the internalized protein is sent for degradation in the lysosomal compartment [12,14].

We and others have shown that slower internalization from the PM depends on the anchoring of wt-CFTR to the actin cytoskeleton [14,15,18,19,20]. This is enabled by the binding of CFTR’s C-terminal postsynaptic density 95, disks large, zonula occludens-1 (PDZ)-binding motif to the second PDZ domain of the Na^+^/H^+^ exchanger regulatory factor (NHERF1) [14,20,21]. NHERF1 has two tandem N-terminal PDZ domains. The first domain (PDZ1) can readily interact with CFTR during its trafficking from the trans-Golgi to the PM [5]. However, the second PDZ domain (PDZ2) is blocked from binding to CFTR by an intramolecular NHERF1 interaction with its C-terminal Ezrin binding domain (EBD) [21], which is only released upon binding to active Ezrin at the PM [14,22]. Ezrin binding to NHERF1 EBD domain then allows NHERF1 PDZ2 to bind CFTR, resulting in its anchoring to the actin cytoskeleton via Ezrin and prolonged apical PM localization and efficient activation of wt-CFTR [14,18,19,23]. Although rF508del-CFTR can also bind to the NHERF1 PDZ1 domain, it fails to recruit Ezrin and switch to NHERF1’s PDZ2, thus lacking anchorage to actin filaments unless coaxed by active Ezrin overexpression [14,19]. In order to understand the underlying molecular mechanism, we recently used mass spectrometry (MS) to analyze rF508del-CFTR-containing complexes selectively immunoprecipitated from the PM of airway cells [24]. We found these PM-located complexes were enriched in Calpain-1, a calcium-dependent protease that has Ezrin among its substrates. Calpain-1 prevented Ezrin recruitment to rF508del-CFTR but not to wt-CFTR, and its depletion or chemical inhibition significantly increased the functional abundance and stability of the rescued mutant channels at the cell surface [24]. However, our MS analysis identified other candidate proteins as potential interactors with rF508del-CFTR at the PM, suggesting additional mechanisms regulating the channel’s decreased stability at the cell surface. Among these proteins was the Yamaguchi sarcoma viral oncogene homolog 1 (YES1) [24], a nonreceptor tyrosine kinase of the Src family, whose dysregulation has been associated with multiple cancer signaling pathways [25]. Here, we describe that YES1 associates at the PM with rF508del-CFTR, but not with wt-CFTR. This is accomplished through the NHERF1-binding adaptor protein YAP1 (YES-associated protein 1), which phosphorylates the Src homology 2 domain-containing transforming protein 1 (SHC1). SHC1 is an adaptor protein that serves as the molecular link between CFTR internalization and the activation of the mitogen-activated protein kinase (MAPK) pathway.

## 2. Materials and Methods

### 2.1. Cell Culture, Treatment and Transfection

CFBE41o- cells were generated by Adv Bioscience LLC and were stably transduced with lentivirus encoding mCherry-Flag-tagged, wt-, or F508del-CFTR under the Tet-ON promoter, as previously described [26]. A clone of CFBE cells stably expressing nontagged F508del-CFTR [27] was selected to stably co-express the halide sensor YFP-F46L/H148Q/I152L (HS-YFP; kindly provided by P. Haggie, University of California, San Francisco, CA, USA, School of Medicine) [14].

All cell lines were cultured at 37 °C with 5% CO_2_ and regularly checked for the absence of mycoplasm infection. They were maintained in minimum essential medium supplemented with GlutaMAX, Earle’s salts, 10% (*v*/*v*) heat-inactivated fetal bovine serum (FBS), and 2 μg/mL puromycin (all from Thermo Fisher Scientific, Waltham, MA, USA). CFBE mCherry-Flag-CFTR cells were additionally selected with 10 µg/mL blasticidin (Invivogen, San Diego, CA, USA), whereas the cells expressing HS-YFP were selected with 400 µg/mL hygromycin B (Thermo Fisher Scientific). In addition, CFTR construct expression in mCherry-Flag-tagged cells was induced with 1 μg/mL doxycycline (Dox; Sigma-Aldrich, Saint Louis, MO, USA).

The cells were reverse transfected as described in [28] with Lipofectamine 2000 (Thermo Fisher Scientific, Waltham, MA, USA) in 35-mm or 60-mm dishes with 200 pmol or 400 pmol of the indicated siRNAs, respectively, and analyzed at the indicated time points. An siRNA oligonucleotide against luciferase (siCtrl, 5′-CGUACGCGGAAUACUUCGA) from Eurofins Genomics was used as a mock control, and the siRNAs used to deplete YES1 (sc-29860), YAP1 (sc-38637) or SHC1 (sc-29480) were commercial mixes (each composed of three different oligonucleotides) from Santa Cruz Biotechnology (Dallas, TX, USA). For ectopic expression of the constitutively active mutant construct pRK5-Myc-H-RAS-V12 [29], a total amount of 6 μg of DNA plasmid (supplemented with empty vector when required) was transfected, with a ratio of 1:3 (μg/μL) of DNA:Lipofectamine per 60-mm dish.

Stock solutions of VX-661 (10 mM, Achemblock, Hayward, CA, USA), SU6656 (10 mM, Sigma-Aldrich), P505-15 (10 mM, Selleckchem, Houston, TX, USA), Selumetinib (10 mM, Santa Cruz Biotechnology, Dallas, TX, USA), Forskolin (Fsk, 10 mM, Sigma-Aldrich, Saint Louis, MO, USA), Genistein (Gen, 50 mM, Sigma-Aldrich), or CFTR-specific inhibitor 172 (inh172, 10 mM, Santa Cruz Biotechnology, Dallas, TX, USA) were prepared at least 1000-fold in DMSO. This ensured that the DMSO concentration during cell treatment did not exceed 0.1% (*v*/*v*).

### 2.2. Protein Thermal Destabilization Assay (Thermal Shift Assay—TS)

CFBE cells expressing either mCherry-Flag-tagged (previously induced with Dox) or untagged F508del-CFTR were incubated for 48 h at 30 °C with VX-661 (5 μM, Achemblock, Hayward, CA, USA). These conditions attenuated deficient protein folding and were previously optimized to achieve rF508del-CFTR PM levels close to those of wt-CFTR [14,24]. The cells were next transferred to 37 °C for 3 h, which restored misfolding and destabilized the rF508del-CFTR at the PM, as described in [24,30].

Cells were then placed on ice, washed three times with ice-cold PBS-CM (PBS (pH 8.0), containing 0.9 mM CaCl_2_ and 0.5 mM MgCl_2_), and left for 5 min in cold PBS-CM. The cells were then analyzed using confocal immunofluorescence, biotinylation of surface proteins, or immunoblotting, as indicated below.

### 2.3. Immunoblotting and Immunofluorescence

The samples were analyzed using immunoblotting as previously described [24,30,31]. The antibodies used for Western blot (WB) were as follows: mouse anti-CFTR clone 596 (obtained through the UNC CFTR antibody distribution program sponsored by Cystic Fibrosis Foundation Therapeutics—CFFT, Bethesda, MD, USA); mouse anti-α-Tubulin clone B-5-1-2 (T5168), mouse anti-Myc clone 9E10 (M5546), mouse anti-phospho-ERK 1/2 (M8159) and rabbit anti-ERK1/2 (M5670) from Sigma-Aldrich; rabbit anti-YES (#65890), and rabbit anti-YAP (#14074) from Cell Signaling; mouse anti-YES (sc-46674), mouse anti-phospho SHC (sc-81519) and mouse anti-SHC (sc-967) from Santa Cruz Biotechnology (Dallas, TX, USA); and rabbit anti-Glut (Ab652) from Abcam. The primary antibodies were detected using secondary, peroxidase-conjugated antibodies (Bio-Rad, Hercules, CA, USA) followed by chemiluminescence detection. For densitometric analysis of WB bands, x-ray films from at least three independent experiments were digitalized, and the images were analyzed using ImageJ 1.53q software (NIH, Bethesda, MD, USA).

For the immunofluorescence analysis, mCherry-Flag-F508del-CFTR CFBE cells were grown on glass coverslips (10 × 10 mm), transfected, induced with Dox (1 μg/mL), and treated as indicated. Next, the cells were rinsed on ice with cold PBS-CM (PBS (pH 8.0), containing 0.9 mM CaCl_2_ and 0.5 mM MgCl_2_) three times and incubated with anti-Flag M2 Ab (F3165, Sigma-Aldrich, Saint Louis, MO, USA) in PBS-CM + 1% (*w*/*v*) BSA (Sigma-Aldrich, Saint Louis, MO, USA) for 90 min at 4 °C without permeabilization. Afterwards, the cells were washed for 5 min with ice-cold PBS-CM under soft agitation for three times and incubated with anti-mouse Alexa Fluor 488 secondary Ab (Thermo Fisher Scientific, Waltham, MA, USA) in PBS-CM + 1% (*w*/*v*) BSA for 30 min at 4 °C. Then, the cells were washed three times with ice-cold PBS-CM and fixed with 4% formaldehyde for 20 min, followed by three washes with PBS + 0.01% Triton X-100. Finally, the cells were stained with 4′,6-diamidino-2-phenylindole (DAPI) and mounted on microscope slides with Vectashield (Vector Laboratories, Newark, CA, USA). Images were acquired on a Leica TCS-SPE confocal microscope.

### 2.4. Biotinylation of Cell Surface Proteins and Internalization Protocol

CFBE cells expressing untagged F508del-CFTR were treated and incubated at the indicated conditions. Afterwards, the cells were placed on ice, rinsed three times with ice-cold PBS-CM (PBS (pH 8.0), containing 0.9 mM CaCl_2_ and 0.5 mM MgCl_2_), and left for 5 min in ice-cold PBS-CM to ensure the arrest of endocytic traffic. To label cell surface proteins, the cells were incubated for 30 min with 0.5 mg/mL of EZ-Link Sulfo-NHS-SS-Biotin (Santa Cruz Biotechnology, Dallas, TX, USA) in PBS-CM. Subsequently, the cells were washed once with PBS-CM and once with ice-cold Tris-Q (100 mM Tris-HCl (pH 7.5), 150 mM NaCl, 0.9 mM CaCl_2_, 0.5 mM MgCl_2_, 10 mM glycine, 1% (*w*/*v*) BSA), then left for 15 min on ice with Tris-Q to quench the reaction. Then, the cells were washed three times with cold PBS-CM and underwent one of the two following steps:(1)*Cell surface protein analysis*: The cells were lysed in a pull-down (PD) buffer (50 mM Tris-HCl (pH 7.5), 100 mM NaCl, 10% (*v*/*v*) glycerol, 1% (*v*/*v*) Nonidet P-40, 0.1% (*v*/*v*) SDS, protease inhibitor cocktail (Sigma-Aldrich, Saint Louis, MO, USA), phosphatase inhibitor cocktail C (Santa Cruz Biotechnology, Dallas, TX, USA)). The cell lysates were cleared at 10,000× *g* for 5 min at 4 °C, and an aliquot was removed while the remaining supernatant was added to streptavidin-agarose beads (Sigma-Aldrich, Saint Louis, MO, USA) that were previously blocked for 1 h with 2% (*w*/*v*) milk in PD-buffer to represent the lysate input. The lysate and beads were rotated for 1 h at 4 °C, centrifuged for 1 min at 3000× *g*, and washed five times with an excess of wash buffer (100 mM Tris-HCl (pH 7.5), 300 mM NaCl, 1% (*v*/*v*) Triton X-100). The captured proteins were eluted in 2× Laemmli buffer with 100 mM of DTT and analyzed using immunoblotting, as described above.(2)*Internalization studies*: Biotin-labeled cells were incubated for 3 h at 37 °C with warm media containing the indicated treatments. Next, the cells were replaced on ice, rinsed with ice-cold PBS-CM, and left for 5 min to block endocytosis. After two ice-cold PBS-CM washes, the cells were incubated for 30 min (2 × 15 min) with ice-cold stripping buffer (60 mM glutathione, 90 mM NaCl, 0.9 mM CaCl_2_, 0.5 mM MgCl_2_, 90 mM NaOH, 10% (*v*/*v*) FBS). The cells were then washed three times with ice-cold PBS-CM and processed as in step 1.

### 2.5. Co-Immunoprecipitation of Membrane CFTR-Associated Complexes

mCherry-Flag-wt and mCherry-Flag-F508del-CFTR CFBE cells were used to co-immunoprecipitate membrane CFTR-associated complexes, as described in [24]. Briefly, after Dox-induced CFTR expression (1 μg/mL, Sigma-Aldrich, Saint Louis, MO, USA), wt-CFTR cells were incubated for 48 h at 37 °C, whereas F508del-CFTR were incubated for 48 h at 30 °C with VX-661-mediated pharmacological correction (5 μM, Achemblock, Hayward, CA, USA). The cells were then placed on ice and incubated under mild agitation for 2 h with anti-Flag M2 Ab (F3165, Sigma-Aldrich, Saint Louis, MO, USA) or a non-specific IgG (anti-HA, H6908, Sigma-Aldrich, Saint Louis, MO, USA) in PBS-CM (PBS (pH 8.0), containing 0.9 mM CaCl_2_ and 0.5 mM MgCl_2_), washed thrice with PBS-CM, and lysed with lysis buffer (50 mM Tris-HCl (pH 7.5), 2 mm MgCl_2_, 100 mm NaCl, 10% (*v*/*v*) glycerol, 1% (*v*/*v*) Nonidet P-40, 0.01% (*v*/*v*) SDS, protease inhibitor cocktail (Sigma-Aldrich, Saint Louis, MO, USA)). The cell lysates were cleared using centrifugation, an input control aliquot collected and the remaining supernatant was cleared using streptavidin–agarose beads (Sigma-Aldrich, Saint Louis, MO, USA). The protein complexes were captured with protein G magnetic beads (Thermo Fisher Scientific, Waltham, MA, USA), washed five times with wash buffer (50 mM Tris-HCl (pH 7.5), 2 mm MgCl_2_, 200 mm NaCl, 1% (*v*/*v*) Nonidet P-40, protease inhibitor cocktail (Sigma-Aldrich, Saint Louis, MO, USA)), eluted in 2× Laemmli buffer with 100 mM DTT, and analyzed through immunoblotting, as described above, with the indicated antibodies.

### 2.6. Co-Immunoprecipitation of Membrane Coaxed rF508del-CFTR

Untagged F508del-CFTR-expressing CFBE cells were reverse transfected as indicated and incubated for 48 h at 30 °C with VX-661 (5 μM, Achemblock, Hayward, CA, USA). The cells were then placed on ice, washed three times with ice-cold PBS-CM (PBS (pH 8.0), containing 0.9 mM CaCl_2_ and 0.5 mM MgCl_2_), and lysed with lysis buffer (50 mM Tris-HCl (pH 7.5), 2 mm MgCl_2_, 100 mm NaCl, 1.5% (*v*/*v*) Nonidet P-40, protease inhibitor cocktail (Sigma-Aldrich, Saint Louis, MO, USA)). The cell lysates were cleared using centrifugation at 10,000× *g* for 5 min at 4 °C, an input control aliquot corresponding to 1/10 of the volume from each condition was removed, and the remaining supernatant was cleared using streptavidin-agarose beads (Sigma-Aldrich) for 1 h with agitation at 4 °C. Next, the cleared lysates were incubated overnight with 500 ng of control Ab (anti-HA, H6908, Sigma-Aldrich, Saint Louis, MO, USA) or anti-YES (#65890, Cell Signaling) at 4 °C with agitation. Then, protein-antibody complexes were incubated with protein G magnetic beads (Thermo Fisher Scientific, Waltham, MA, USA) for 1 h at 4 °C with agitation, and after five washes with wash buffer (50 mM Tris-HCl (pH 7.5), 2 mm MgCl_2_, 200 mm NaCl, 1% (*v*/*v*) Nonidet P-40, protease inhibitor cocktail (Sigma-Aldrich, Saint Louis, MO, USA)), the captured proteins were eluted in 2× Laemmli buffer with 100 mM DTT and analyzed using immunoblotting, as described above, with the indicated antibodies.

### 2.7. CFTR Functional Assay by Halide-Sensitive YFP (HS-YFP)

CFTR activity was determined using the above-mentioned F508del-CFTR and HS-YFP-expressing CFBE cells, as described in [14,30]. Briefly, the cells were treated and/or transfected as indicated, then washed with PBS and incubated for 30 min in PBS-containing compounds for CFTR stimulation/inhibition (5 µM Fsk, 50 µM Gen, or 25 µM inh172). The HS-YFP fluorescence decay in cells was then analyzed by recording fluorescence continuously (500 ms/point) for 10 s (baseline) and then for 40 s after the rapid (<1 s) apical addition of isomolar PBS in which 137 mM Cl^−^ was replaced by I^−^ (PBSI, final NaI concentration in the well: 100 mM). After background subtraction, the cell fluorescence recordings were normalized for the initial average value measured before the addition of I^−^. The initial rate of fluorescence decay (QR), an indicator of the rate of halide transport by CFTR [27], was derived by fitting the curves to an exponential decay function using GraphPad 5.0.

### 2.8. Statistical Analysis

Quantitative results are shown as means ± SEM of at least three replicate observations. We used Student’s *t* tests to compare paired sets of data and one-way ANOVAs followed by Tukey’s posttests for multiple data sets. Differences were considered significant when *p* < 0.05.

## 3. Results

### 3.1. Interaction with YES1 Decreases rF508del-CFTR Functional Permanence at the PM

In our previous work, we determined that tyrosine kinase YES1 was part of the protein complexes interacting with corrector-rescued F508del-CFTR at the PM of bronchial epithelial airway cells using MS [24]. For this study, we used CFBE41o- cells stably expressing either mCherry-wt-CFTR or mCherry-F508del-CFTR with an extracellular Flag-tag, allowing us to selectively co-immunoprecipitate CFTR-containing protein complexes from the PM of intact cells. As shown in Figure 1A, YES1 was detected in PM-derived co-precipitates from VX-661-rescued (5 µM for 48 h) F508del-CFTR (rF508del-CFTR). However, it was absent from equivalent precipitates captured from the PM of mCherry-wt-CFTR cells. For functional validation, we first used the Flag-tag to immunostain the CFTR at the PM of intact cells after depleting the endogenous YES1 levels by 80% using validated commercial siRNAs (Figure 1B). We observed a clear increase in the abundance of rF508del protein at the cell surface without any noticeable change in the overall CFTR protein abundance, as assessed using the intracellular mCherry tag (Figure 1C). Furthermore, a similar increase in cell surface rF508del protein was also apparent following a 3 h treatment with 10 µM SU6656, a chemical YES1 kinase inhibitor (Figure 1C). Finally, we tested for changes in CFTR-mediated ion transport using CFBE cells co-expressing untagged F508del-CFTR and the halide sensor YFP-F46L/H148Q/I152L (HS-YFP) [24]. We observed an over 2-fold increase in the rate of CFTR-mediated ion transport when YES1 levels were either depleted through RNA interference, or when its kinase activity was inhibited with SU6656 in VX-661-treated cells, consistent with the observed increased immunostaining signal of rF508del-CFTR at the cell surface (Figure 1D,E).

### 3.2. Inhibition of YES1 Impairs Thermal Destabilization and Internalization of PM-Bound rF508del-CFTR

We next investigated whether the increase in the PM abundance of rF508del-CFTR upon YES1 downregulation reflected increased retention of the protein at the airway cell’s surface. For this, we employed the thermal shift (TS) assays described in [24,30], where rF508del-CFTR at the cell surface was first coaxed to accumulate through thermal stabilization of its protein folding by incubating the cells at 30 °C. Subsequently, the cells were placed at 37 °C for 3 h to induce rF508del-CFTR thermal destabilization, leading to its internalization [24,30]. These conditions allowed us to assess the effect of YES inhibitors on the channel’s stability at the cell surface in comparison with the mock treatment. In two complementary assays (see Figure 2A), the cell surface rF508del-CFTR levels were assessed using protein biotinylation. While one assay detected rF508del-CFTR remaining at the PM after TS, another isolated the biotinylated rF508del-CFTR protein that had been internalized upon TS both in the absence and presence of YES1 inhibitors SU6656 (10 µM) or P505-15 (1 µM) (Figure 2B,C). We observed that YES1 inhibition significantly prevented rF508del-CFTR internalization upon thermal destabilization, allowing most of the rescued protein to remain at the PM (Figure 2B, quantified in Figure 2C).

### 3.3. The Interaction between YES1 and rF508del-CFTR Is Mediated by YAP1

Although we previously identified YES1 as a constituent of the rF508del-CFTR protein complex at the PM [24], when applying the STRING protein–protein interaction (PPI) network algorithm (https://string-db.org, accessed on 3 May 2021), it became clear that the strength of the annotated evidence for a direct interaction between YES1 and the NHERF1/Ezrin/CFTR complex was weaker than that of other complex components (Figure 3A). However, using the STRING algorithm network expansion function, we identified YES-associated protein 1 (YAP1), as a strong interactor of YES1 with documented evidence of association with NHERF1 (Figure 3B). YAP1 is a transcription co-regulator in the Hippo signaling pathway, which is involved in cell proliferation, apoptosis, and various stress responses [32]. However, YAP1 was first identified as an adaptor protein that associates with the SH3 domain of YES1 kinase, modulating its activity and PM localization through a direct PDZ-mediated interaction with NHERF1 [33,34]. We therefore probed CFTR-containing complexes immunoprecipitated from the PM of mCherry-Flag-rF508del- and mCherry-Flag-wt-CFTR-expressing CFBE cells with anti-YAP1 antibodies. As observed for YES1 in Figure 1A, we determined that YAP1 co-precipitated only with rF508del-CFTR (Figure 3C). In order to understand the role of YAP1 in this complex, we tested its requirement for the YES1/F508del-CFTR protein complex. For this, we took CFBE cells stably expressing untagged F508del-CFTR [31] and coaxed most of the channel to the PM through treatment with VX-661 for 48 h at 30 °C. Then, we immunoprecipitated YES1 and were able to confirm the presence of untagged rF508del-CFTR in the co-precipitate (Figure 3D). This assay was then repeated using cells previously depleted of endogenous YAP1 expression. Depletion of YAP1 in these cells reached ~70% (*p* < 0.01) and was sufficient to prevent rF508del-CFTR from co-precipitating with YES1 (Figure 3D), indicating that YAP1 mediated the interaction. Further supporting this observation, the siRNA-mediated downregulation of YAP1 significantly increased the thermal stability of rF508del-CFTR at the PM to levels similar to those achieved through YES1 depletion (Figure 3E, quantified in Figure 3F).

### 3.4. YES1 Participates in the Removal of rF508del-CFTR from the PM via the MEK/ERK1/2 MAPK Pathway

So far, our data suggested that a YES1/YAP1 interaction with F508del-CFTR promotes its internalization from the PM. It was previously described that the activation of the mitogen-activated protein kinase (MEK)/extracellular signal-regulated kinase 1/2 (ERK 1/2) mitogen-activated protein kinase (MAPK) pathway, known to regulate cell proliferation and survival, also plays a key role in triggering the internalization of wt-CFTR from the PM of human airway cells [35]. To investigate whether the same mechanism could be involved in the effect of the YES1/YAP1 complex, we proceeded to block MAPK signaling in CFBE cells using the MEK-selective inhibitor selumetinib (10 µM for 3 h) and determined the effect on the thermal destabilization of rF508del-CFTR at the PM. Selumetinib treatment produced a significant, over 80% (*p* < 0.001) decrease in MAPK activity, as measured by the phosphorylation levels of ERK1/2 at Thr202 and Tyr204 (Figure 4A), the downstream effector kinase in this pathway [36]. Importantly, we observed that the inhibition of MAPK signaling produced a significant retention of rF508del-CFTR at the PM upon thermal destabilization (Figure 4A, quantified in Figure 4B), comparable to the extent induced by YES1 inhibition or YAP1 depletion (see Figure 3E,F).

To determine if the two events were connected, we inhibited YES1 activity with SU6656, as before, in cells expressing a constitutively active H-RAS mutant (H-RAS-V12) that was able to fully stimulate MAPK signaling, bypassing the need for receptor activation at the PM [37]. This experiment showed that while YES1 inhibition in mock transfected cells led to a significant retention of rF508del-CFTR at the PM after thermal destabilization, the effect was abrogated in cells expressing H-RAS-V12 (Figure 4C, quantified in Figure 4D). Noteworthy, SU6656 treatment was sufficient to reduce ERK1/2 phosphorylation by over 70% (*p* < 0.001) in mock transfected CFBE cells, but not in cells expressing H-RAS-V12 (Figure 4C). These data indicate that YES1 interacts with the MAPK pathway activity upstream of RAS, and that the effect of YES1 on rF508del-CFTR PM retention is mechanistically connected with MAPK pathway activity.

### 3.5. Phosphorylation of SHC1 by YES1 Links rF508del-CFTR Internalization to MAPK Pathway Activation

In order to identify a direct link between YES1 activity and MAPK activation, we investigated SHC-transforming protein 1 (SHC1), which has been reported to be phosphorylated by YES1 at Tyr239 and Tyr240 [38,39]. Phosphorylated SHC1 acts as an adaptor protein that improves the signal transduction between stimulated PM receptors and RAS proteins, leading to the stimulation of the MAPK cascade [38,39,40]. While we could not reliably detect SHC1 co-precipitating with YES1 in our experimental settings (possibly reflecting a transient enzyme–substrate interaction), we could nevertheless detect a significant (*p* < 0.01) decrease in SHC1 Tyr239/240 phosphorylation upon inhibition of YES1 activity after 3 h treatment with either SU6656 (10 µM) or P505-15 (1 µM) (Figure 5A). Moreover, an over 70% depletion of endogenous SHC1 levels (*p* < 0.01) in CFBE cells led to a significant (*p* < 0.01) retention of rF508del-CFTR at the PM to levels comparable to those observed after YES1 or YAP1 downregulation upon thermal destabilization (Figure 5B, quantified in Figure 5C).

Taken together, these results suggest that SHC1 phosphorylation by YES1 mediates rF508del-CFTR internalization via MAPK signaling. Consistently, depletion of SHC1 in these cells also reduced ERK1/2 phosphorylation (*p* < 0.01) to levels comparable to those of YES1 inhibition (Figure 5A,B).

## 4. Discussion

Despite the significant progress that CFTR modulators have brought to CF therapy, their clinical effects remain limited by their inability to fully restore rF508del-CFTR stability at the PM [3,10,11,14]. Hence, recent efforts have been made to better characterize the molecular mechanisms of CFTR proteostasis in order to identify targetable molecules that can modulate these processes and enhance the effectiveness of CFTR modulator therapy [3,41]. In previous work, we characterized a novel pathway that regulates wt-CFTR retention at the PM: when phosphorylated by spleen tyrosine kinase (SYK) at its Tyr512 residue, wt-CFTR is removed from the PM [42,43]. The effect was found to be mediated by adaptor protein SHC1, which recognizes Tyr512-phosphorylated CFTR through its phosphotyrosine-binding domain and links CFTR internalization to the activation of the MAPK pathway [35,43,44] (see Figure 6A). In contrast, rF508del-CFTR internalization did not respond to SYK modulation, likely due to the misfolding caused by the Phe508 deletion, which makes the kinase fail to recognize and phosphorylate the nearby Tyr512 residue [42,43]. However, our findings presented here uncovered an alternative mechanism that links rF508del-CFTR to SHC1 and MAPK pathway-associated internalization. We showed that rF508del-CFTR (but not wt-CFTR) at the PM binds to YES1 kinase through the adaptor protein YAP1. YAP1 has been described to bridge the interaction of YES1 with NHERF1 through a strong interaction between YAP1′s C-terminus and NHERF1’s second PDZ domain, even in the absence of Ezrin [34]. This is consistent with our previous finding that most rF508del-CFTR at the PM remains bound to NHERF1’s first PDZ [14]. This results from the interaction of rF508del-CFTR with the protease Calpain-1 at the PM [24], which prevents Ezrin recruitment and consequently blocks rF508del-CFTR from switching to NHERF1’s PDZ2 [14,24] (see Figure 6B). 

In addition, we demonstrated that rF508del-CFTR-bound YES1 can also phosphorylate SHC1. Consistently, interfering with either component of this complex or inhibiting YES1 activity resulted in decreased SHC1 phosphorylation and a significant delay in rF508del-CFTR internalization. We also showed that overexpression of a constitutively active mutant RAS protein (H-RAS-V12) could bypass SHC1-mediated activation of the MAPK pathway and lead to rF508del-CFTR internalization. MAPK pathway activation occurs downstream of receptor tyrosine kinases (RTK) in response to mitogenic stimuli [45,46]. In response to RTK activation, a complex composed by the GRB2 scaffold protein and the RAS guanine exchange factor SOS1 (which induces RAS activation) is recruited to the activated RTK, either directly or indirectly via adaptors such as SHC1 [45] (see Figure 6). SHC1 contains an N-terminal PTB domain and a C-terminal SH2 domain, which are both able to bind phosphorylated tyrosine residues on other proteins [46]. These flank a central proline-rich region that also contains the tyrosine sites for SHC1 phosphorylation [39]. SHC1 phosphorylation at Tyr239/240 by YES1 greatly increased its affinity to GRB2/SOS1, boosting RAS activation and MAPK signaling [38,39,46]. Thus, in contrast to wt-CFTR, the phosphorylation of SHC1 by YES1 upon its recruitment to rF508del-CFTR complexes at the PM may further contribute to the rapid internalization of the rescued mutant channels by improving MAPK signaling in their vicinity (Figure 6). Determining the precise mechanism by which MAPK activation promotes CFTR internalization will require further investigation.

## 5. Conclusions

Our findings provide new, important insights into the mechanisms regulating the accelerated internalization of rF508del-CFTR in airway cells. Moreover, the involvement of YES1 kinase and the MAPK pathway in removing the rescued channels from the PM opens new avenues for future CF research. In this study, we demonstrated that treatment with the MEK inhibitor selumetinib significantly improved the stability and retention of VX-661-rescued F508del-CFTR at the PM. Similar results were observed upon treatment with the YES1 inhibitors SU6656 and P505-15. Having identified these new pathways, it will now be important to validate these findings in patient-derived materials. It will also be important to determine the extent to which these pathways hinder the effect of recently clinically explored VX-661/VX-445 additive corrector combination and whether the YES1/MAPK-mediated PM removal mechanism can be attenuated by any of the many new CFTR modulators in clinical trials. Moreover, while several drugs are currently available to inhibit both the MAPK pathway and the activity of Src family kinases, such as YES1, [25,47] the continuous inhibition of these pathways may have deleterious side effects [48]. Therefore, translation of our findings into a CF therapeutic context will require additional research in order to develop ways to safely target these pathways to enhance CFTR modulator therapy. Notwithstanding, in our view, our data bring exciting new avenues for further exploration in both CF research and treatment.

## Figures and Tables

**Figure 1 biomolecules-13-00949-f001:**
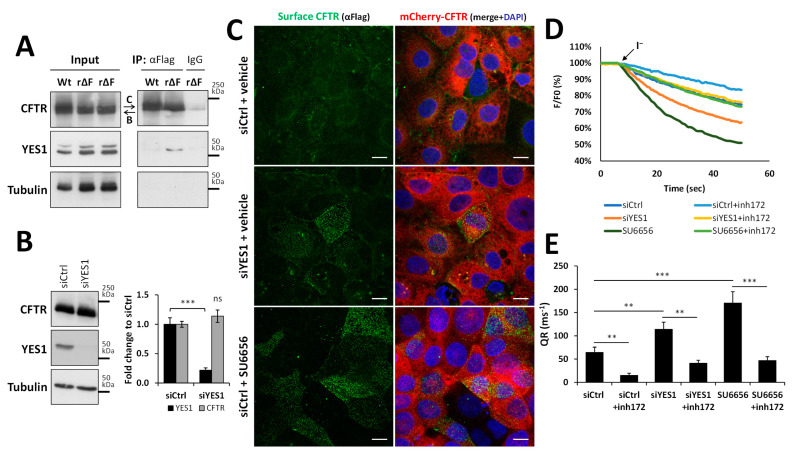
Interaction with YES1 decreases the PM abundance and function of rF508del-CFTR in airway cells. (**A**) Intact CFBE cells expressing extracellularly Flag-tagged mCherry-wt-CFTR (Wt) or mCherry-F508del-CFTR (rΔF, after rescue to the PM by 48 h treatment with 5 µM VX-661), were labelled with either an anti-Flag antibody or a non-specific IgG prior to non-denaturing lysis, and antibody-bound CFTR complexes were precipitated using protein G-coupled magnetic beads. Input protein levels were adjusted so that equivalent amounts of wt- and rF508del-CFTR were precipitated. Both input lysates and co-precipitates were analyzed using WB to assess the levels of CFTR and YES1. Tubulin (known to not interact with CFTR at the PM [24]) was used as an additional co-precipitation control. (**B**) Efficiency of siRNA-mediated depletion in CFBE cells expressing mCherry-F508del-CFTR, rescued as in (**A**), transfected with either a mock siRNA (siCtrl) or a commercial triple oligonucleotide mix against YES1 (siYES1). Representative WBs of CFTR, YES1, and tubulin (used as loading control) are shown (left panels), as well as the quantifications as means ± SEM from four independent experiments (right panel). (**C**) Immunofluorescence images of CFBE cells expressing Flag-tagged mCherry-F508del-CFTR rescued to the PM by 48 h treatment with 5 µM VX-661. The cells were transfected as in (**B**) and treated for 3 h with either vehicle (DMSO) or 10 µM of YES1 inhibitor SU6656. rF508del-CFTR at the surface of intact cells was immunolabelled on ice using an anti-Flag antibody followed by an Alexa 488-conjugated secondary antibody. Cells were then fixed, and the nuclei were stained with DAPI. Confocal images of the cells’ surface, showing surface CFTR staining in green (left panels) and a merged overlay image of surface (green) and total CFTR signals (mCherry, red) along with the stained nuclei (DAPI, blue) are shown in the right panels. White scale bars represent 10 µm. (**D**) Representative traces of ion transport activity measured through iodide-induced HS-YFP sensor fluorescence decay of untagged F508del-CFTR CFBE cells treated with 5 μM VX-661 for 48 h. The cells were transfected and treated as in (**C**) and co-treated with or without 25 μM of inh172 15 min prior to stimulation for 30 min in PBS with Fsk (5 µM) and Gen (10 µM), in the presence or absence of inh172. This was followed by continuous fluorescence recording and the addition of I^-^ (represented by the black arrow, final concentration 100 mM). (**E**) Quantification of HS-YFP fluorescence quenching rates (QR) of at least five independent assays for each condition, calculated by fitting the iodide assay results to exponential decay curves. The means ± SEM are shown. ns—not significant, ** *p* < 0.01, and *** *p* < 0.001 between conditions indicated by the horizontal lines.

**Figure 2 biomolecules-13-00949-f002:**
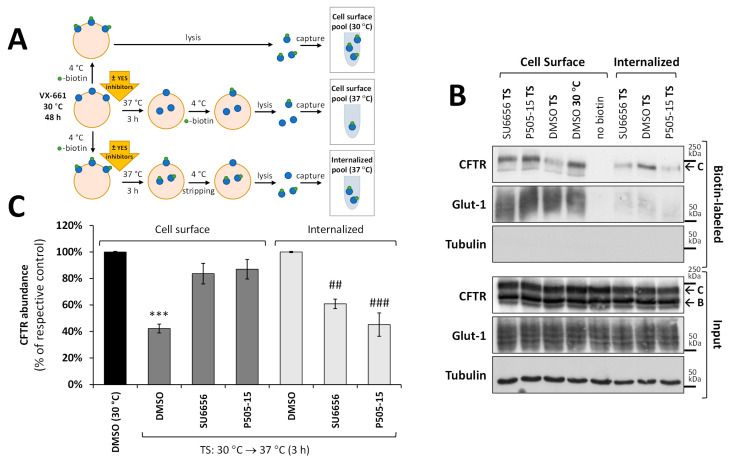
YES1 inhibition increases rF508del-CFTR retention at the PM upon thermal destabilization. (**A**) Diagram depicting the parallel cell surface protein biotinylation assays used to assess rF508del-CFTR thermal stability and internalization. Replicate dishes of CFBE cells expressing F508del-CFTR were incubated for 48 h at 30 °C in the presence of 5 µM of VX-661. One of the replicates was directly placed on ice, then the surface proteins were labeled with sulfo-NHS-SS-biotin, lysed, and the surface-labeled proteins were captured using streptavidin beads. These precipitates represented the input amount of CFTR at the PM without any thermal destabilization (DMSO 30 °C). A second set of replicates were moved to 37 °C in the absence (DMSO) or presence of YES1 inhibitors (10 µM SU6656 or 1 µM P505-15). Three hours later, surface proteins were labelled and isolated, as described above. This second set revealed the amount of CFTR remaining at the PM after thermal destabilization (TS) and inhibitor treatment. A third set of replicates was first labeled with biotin, then placed at 37 °C for 3 h in the presence or absence of YES inhibitors. These samples were then returned to ice, and the labeled CFTR remaining at the cell surface was stripped from biotin with glutathione prior to lysis and isolation of the labeled proteins that entered the cells. This third set represents the amount of CFTR internalized from the surface upon TS and inhibitor treatment. (**B**) Analysis of input lysates and biotin-labelled fractions obtained as described in (**A**). WBs representative of five independent assays, probed with antibodies against the indicated proteins, are shown. Glucose transporter 1 (Glut-1) and tubulin were used as controls for the equivalence and purity of the biotinylated fractions, respectively. (**C**) Quantification of CFTR abundance in the biotinylated fraction (mean ± SEM) in (**B**) after normalization to Glut-1 levels and to the respective controls. *** *p* < 0.001 relative to DMSO (30 °C), ^##^ *p* < 0.01 and ^###^ *p* < 0.001, both relative to DMSO in the internalized set.

**Figure 3 biomolecules-13-00949-f003:**
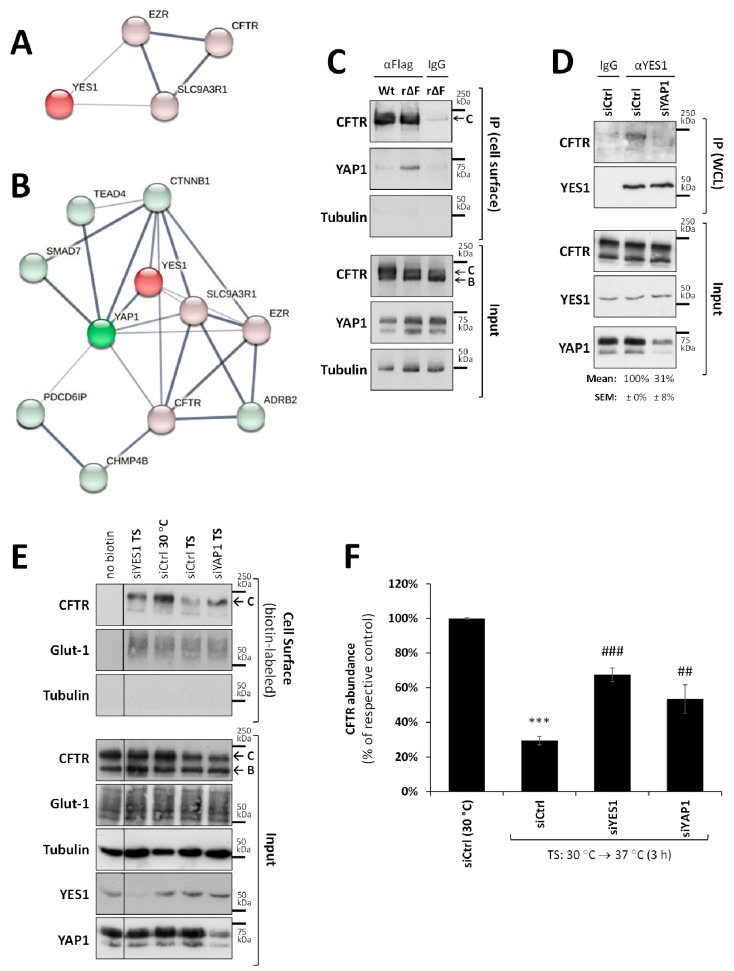
YAP1 is required for the binding of YES1 to rF508del-CFTR complexes at the PM. (**A**) STRING-based analysis (https://string-db.org/, accessed on 19 October 2021) of the strength of annotated evidence on the interaction between YES1 and the CFTR/NHERF1 (SLC9A3R1)/Ezrin (EZR) membrane anchoring complex (the thickness of the grey lines is proportional to the degree of confidence for the interaction between the two proteins they connect, extrapolated from text mining, experimental, and database-collected evidence). (**B**) A STRING-generated expanded interaction network, extended (green nodes) around the core complex in (**A**) (red nodes). (**C**) CFTR-containing complexes were immunoprecipitated from the PM of intact CFBE cells expressing extracellularly Flag-tagged mCherry-wt-CFTR (Wt) or mCherry-F508del-CFTR, as described for Figure 1A. Both input lysates and co-precipitates were analyzed using WB to assess the levels of CFTR and YAP1. Tubulin was used as an additional co-precipitation control. (**D**) CFBE cells expressing untagged F508del-CFTR transfected with either mock siRNA (siCtrl) or a commercial siRNA mix targeting YAP1 (siYAP1), were incubated with 5 µM of VX-661 for 48 h at 30 °C to coax most of the mutant channel to the PM. The cells were then lysed in non-denaturing conditions and YES1 immunoprecipitated with a specific antibody (a non-specific IgG was used as a control) from whole cell lysates (WCL). Both input lysates and co-precipitates were analyzed using WB to assess the levels of precipitated YES1 and co-precipitated rF508del-CFTR. (**E**) Thermal shift (TS) assay, as described in Figure 2, to assess the thermal stability of untagged rF508del-CFTR in CFBE cells transfected with mock siRNA (siCtrl) or one of two commercial siRNA mixes targeting either YAP1 (siYAP1) or YES1 (siYES1). WBs representative of at least four independent assays, probed with antibodies against the indicated proteins, are shown. Glucose transporter 1 (Glut-1) and tubulin were used as controls for the equivalence and purity of the biotinylated fractions, respectively. (**F**) Quantification of CFTR abundance in the biotinylated fraction (mean ± SEM) in (**E**) after normalization to Glut-1 levels and to siCtrl (30 °C). *** *p* < 0.001 relative to siCtrl (30 °C), ^##^ *p* < 0.01 and ^###^ *p* < 0.001, both relative to siCtrl (TS).

**Figure 4 biomolecules-13-00949-f004:**
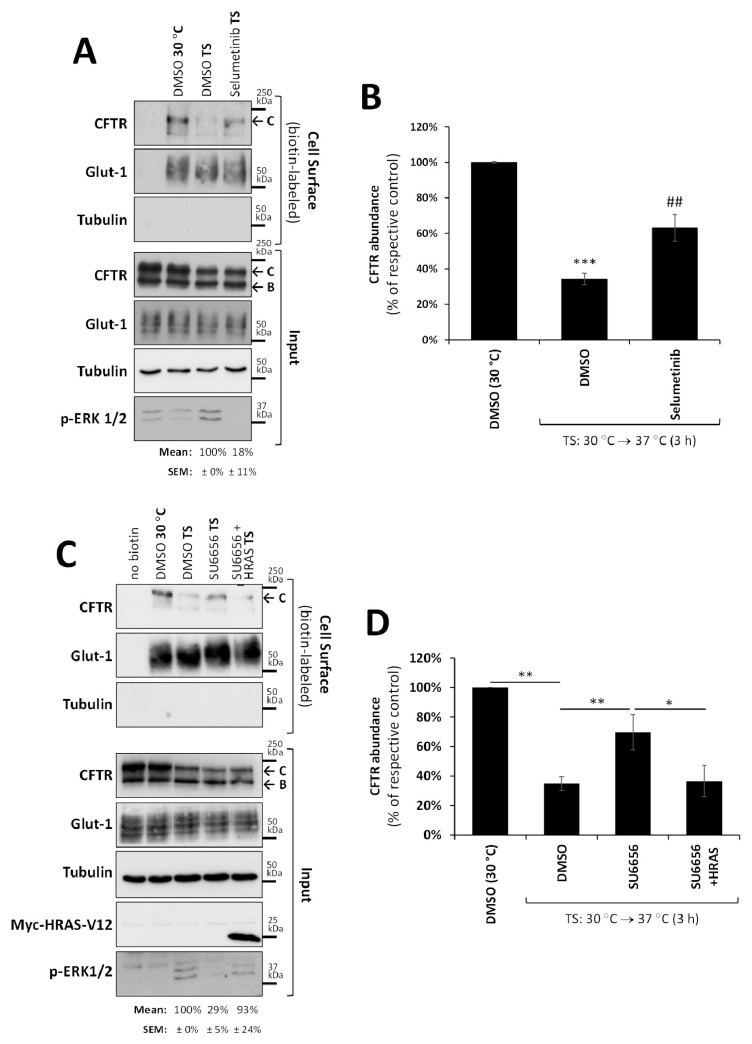
The MAPK pathway participates in YES1-mediated internalization of rF508del-CFTR at the PM. The thermal shift (TS) assay, as described in Figure 2, was used to assess the thermal stability of untagged rF508del-CFTR in CFBE cells. (**A**) The cells were treated for 3 h with either vehicle (DMSO) or 10 µM of selumetinib, or (**C**) transfected with empty vector or Myc-H-RAS-V12 (HRAS) and then treated for 3 h with vehicle (DMSO) or 10 µM of SU6656, as indicated. WBs representative of input and cell surface fractions from four independent assays, probed with antibodies against the indicated proteins, are shown. Glucose transporter 1 (Glut-1) and tubulin were used as controls for the equivalence and purity of the biotinylated fractions, respectively. H-RAS V12 was detected using an anti-Myc antibody, and an anti-phosphorylated ERK1/2 antibody was used to monitor MAPK pathway activity, which was quantified and shown below the respective blot lanes. (**B**,**D**) Corresponding quantification of CFTR abundance in the biotinylated fractions (mean ± SEM) from four independent assays after normalization to Glut-1 levels and DMSO (30 °C). * *p* < 0.05, ** *p* < 0.01, and *** *p* < 0.001, relative to DMSO (30 °C) in (**B**), and as indicated by the horizontal lines in (C); ^##^ *p* < 0.01, relative to DMSO (TS) in (**B**).

**Figure 5 biomolecules-13-00949-f005:**
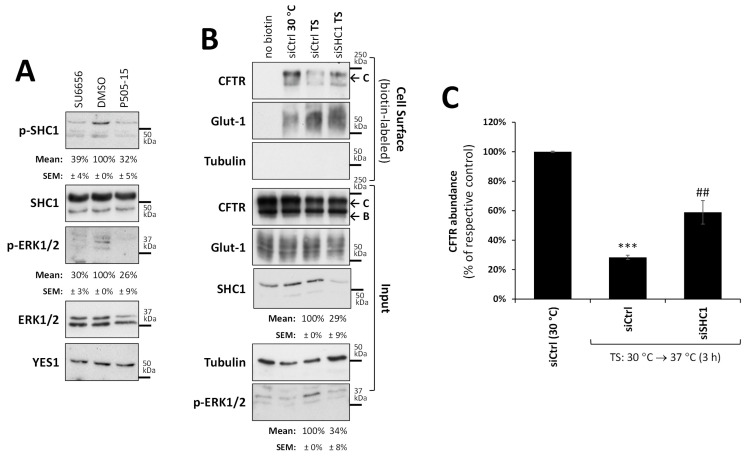
SHC1 phosphorylation by YES1 mediates rF508del-CFTR internalization via MAPK pathway signaling. (**A**) Effects of YES1 inhibitors. Lysates from F508del-CFTR expressing CFBE cells were incubated with 5 µM of VX-661 for 48 h at 30 °C, treated with either vehicle (DMSO), SU6656 (10 µM), or P505-15 (1 µM) for 3 h at 37 °C, then were analyzed using WB. Immunoblots representative of three independent experiments, probed with antibodies against the indicated proteins, are shown. p-SHC1 indicates the level of SHC1 phosphorylation at Tyr239/240 in the different conditions, and an anti-phosphorylated ERK1/2 antibody (p-ERK1/2) was used to monitor MAPK pathway activity (both show quantified band intensities below their respective blots). (**B**) Thermal shift (TS) assay described in Figure 2 to assess how much rF508del-CFTR remained at the PM after thermal destabilization in CFBE cells transfected either with a mock siRNA (siCtrl) or a siRNA against SHC1 (siSHC1). WBs representative of input and cell surface fractions, probed with antibodies against the indicated proteins, are shown. Glucose transporter 1 (Glut-1) and tubulin were used as controls for the equivalence and purity of the biotinylated fractions, respectively. Quantifications of SCH1 depletion efficiency and ERK1/2 phosphorylation levels are shown below their respective blots. (**C**) Corresponding quantification of CFTR abundance in the biotinylated fractions (mean ± SEM) from three independent assays after normalization to Glut-1 levels and to siCtrl (30 °C). *** *p* < 0.001 relative to siCtrl (30 °C), ^##^ *p* < 0.01 relative to siCtrl (TS).

**Figure 6 biomolecules-13-00949-f006:**
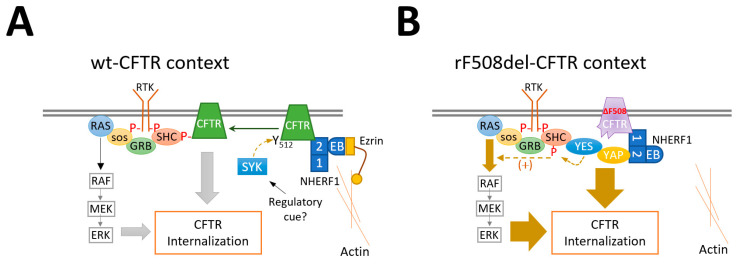
Proposed model for SHC1-mediated removal of CFTR from the PM through activation of the MAPK pathway. (**A**) Phosphorylation of PM-anchored wt-CFTR at Tyr512 by SYK kinase leads to its recognition and binding by the adaptor protein SHC1. This links CFTR internalization to the activation of the MAPK pathway downstream of receptor tyrosine kinases. (**B**) F508del-CFTR pharmacologically rescued to the PM is not phosphorylated by SYK, but its deficient anchoring to the actin cytoskeleton allows its interaction with the YES1 kinase via the adaptor protein YAP1 and the scaffold protein NHERF1. SHC1 is a substrate for YES1 at the PM, and its phosphorylation by YES1 increases its affinity to membrane receptors in the vicinity, enhancing their activation of the MAPK pathway. This could contribute to the much faster internalization rate of rF508del-CFTR compared to the wild-type protein. RTK—receptor tyrosine kinases; EB—Ezrin binding domain; 1-2—NHERF1’s PDZ1 and PDZ2.

## Data Availability

This study did not report any data not shown in the manuscript.

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
