# Peer review of "YES1 Kinase Mediates the Membrane Removal of Rescued F508del-CFTR in Airway Cells by Promoting MAPK Pathway Activation via SHC1"

_biomolecules, 2023, doi:10.3390/biom13060949_

Round 1
Reviewer 1 Report
The paper presents an original and very interesting observation on the role of YES1 protein kinase and its YAP1 adaptor in regulating the stability on the plasma membrane of rescued F508del CFTR. The experimental approach is conducted in rigorous fashion in the CFBE41o- cell line, as expected in a thorough investigation on transmembrane signalling that can be hurdly carried on in primary bronchial epithelial cells. Although it is therefore conceivable that in the first report CF primary cells are not included in the experimental plan, some conclusions on the possibility of applying YES1 or YAP1 inhibitors by aerosol are frankly too ahead of a safe an cautios application of molecules that are expected to have a broad effect on cell biology. The reviewer strongly suggest to focus conclusions more on the mechanistic advancement provided by the results than on triumphalist therapeutic applications in pwCF.
Author Response
Response to Reviewer 1
The paper presents an original and very interesting observation on the role of YES1 protein kinase and its YAP1 adaptor in regulating the stability on the plasma membrane of rescued F508del CFTR. The experimental approach is conducted in rigorous fashion in the CFBE41o- cell line, as expected in a thorough investigation on transmembrane signalling that can be hurdly carried on in primary bronchial epithelial cells. Although it is therefore conceivable that in the first report CF primary cells are not included in the experimental plan, some conclusions on the possibility of applying YES1 or YAP1 inhibitors by aerosol are frankly too ahead of a safe an cautios application of molecules that are expected to have a broad effect on cell biology. The reviewer strongly suggest to focus conclusions more on the mechanistic advancement provided by the results than on triumphalist therapeutic applications in pwCF.
The authors thank the reviewer for appreciative comments and constructive criticism of our work. We acknowledge that our enthusiasm with the findings may have led us to overstate their therapeutic potential. We have toned-down that aspect in the conclusions section, in accordance.
Reviewer 2 Report
This is an interesting study well conducted with good results and interpretation. However, several important points required attention:
1. authors used a very artificial cell model; CFBE cells expressing cherry-flagged CFTR and cotransfected for YFP mesurement. The proteins such as YES, YAP, MAPK... are involved in intracellular pathways that might interfere with cell stress, proliferation, survival and so on. Would it be possible to recap some of the key experiments using human airway epithelial cells to verify that the same results are obtained?
2. the function of CFTR is studied here only by the fluorescence of the YFP sensor. I would prefer the Ussing chamber assay instead, the gold standard for detecting F508del-CFTR activity. The YFP method, also useful to some extent for measuring ion flux, is very sensitive but not always reproducible when recording transepithelial current. CFBE cells are easy to culture for Ussing chamber recording purposes
3. why using Vx661 instead of the triple combinaison ETI? Matos et al (ref 24) used Vx809 and both are correctors type 1. Elexacaftor is a corrector of a different nature which could modify the structure of the CFTR and therefore the interaction with the molecular partners differently. In addition Ivacaftor, the potentiator, is also known to alter the stability of F508del-CFTR. In the presence of these modulators, what would be the effects of YES silencing or inhibition?
Quality is fine
Author Response
Response to Reviewer 2
This is an interesting study well conducted with good results and interpretation. However, several important points required attention:
The authors thank the reviewer for appreciative comments and constructive criticism of our work.
- authors used a very artificial cell model; CFBE cells expressing cherry-flagged CFTR and co-transfected for YFP mesurement. The proteins such as YES, YAP, MAPK... are involved in intracellular pathways that might interfere with cell stress, proliferation, survival and so on. Would it be possible to recap some of the key experiments using human airway epithelial cells to verify that the same results are obtained?
The authors acknowledge the relevance of the reviewer’s comment. We are confident that the reviewer can understand that the amount of experimental manipulation required for the characterization of the molecular mechanisms we describe would not be possible in patient-derived materials. However, we do envision collaboration with groups with access to these material to further validate our findings. Unfortunately, these experiments will not be possible within a reasonable timeframe, much less within the 10-day response time allotted by the journal.
Notwithstanding, we would like to point out that we did use two different CFBE clones in our experiments - one expressing cherry-flagged CFTR and the other expressing the untagged channel, and both gave equivalent results. In addition, in accordance with the reviewer’s concern, we amended the conclusions section, toning down the direct therapeutic applicability of our findings and highlighting the need for further research, namely to verify our observations in patient-derived material and determine the impact of other CFTR modulator drugs on the described mechanisms.
- The function of CFTR is studied here only by the fluorescence of the YFP sensor. I would prefer the Ussing chamber assay instead, the gold standard for detecting F508del-CFTR activity. The YFP method, also useful to some extent for measuring ion flux, is very sensitive but not always reproducible when recording transepithelial current. CFBE cells are easy to culture for Ussing chamber recording purposes.
The authors fully agree with the reviewer that electrophysiological techniques such as Ussing chamber assays are a more accurate approach for measuring CFTR activity. However, we believe that the YFP-based assay was the right approach to assess the impact of our candidate proteins using siRNAs, as the latter do not transfect efficiently in polarized cells. We stress that our purpose at that particular point was not to get a precise measurement of rescued CFTR function, but simply to monitor for positive changes in CFTR-mediated ion transport. Please note that we were always careful to use a CFTR inhibitor (inh172) to exclude any non-CFTR related effects. We hope the reviewer can acknowledge that, for these purposes and in these conditions, the YFP sensor assay is sufficient.
- why using Vx661 instead of the triple combinaison ETI? Matos et al (ref 24) used Vx809 and both are correctors type 1. Elexacaftor is a corrector of a different nature which could modify the structure of the CFTR and therefore the interaction with the molecular partners differently. In addition Ivacaftor, the potentiator, is also known to alter the stability of F508del-CFTR. In the presence of these modulators, what would be the effects of YES silencing or inhibition?
As the reviewer correctly mentioned, in our initial proteomics study (ref.24) we used VX-809 to rescue F508del-CFTR to the PM. In the meantime, we determined that VX-661, developed based on VX-809 structure, had much less impact on CFBE cell viability (10.3389/fmolb.2021.812101) and observed that this was particularly relevant for cells undergoing transient transfection. Hence, we confirmed the co-precipitation of YES1 with VX-661-rescued F508del-CFTR at the PM and proceeded to use VX-661 in this study.
The reason for not adding additional modulators was precisely to maintain the original conditions so as to characterize the mechanism behind CFTR upregulation at the PM upon YES1 depletion. Nonetheless, we agree with the reviewer that it will be important in the future to determine if the removal of rescued CFTR from the PM via the YES1/MAPK mechanism can be overcome or attenuated by other CFTR modulators, including elexacaftor and the many others in clinical trial. In accordance, we have included reference to this point in the revised conclusions section.
Round 2
Reviewer 2 Report
Although some experiments are missing, i agree that this is a preliminary study before exploring human airway epitheliel cells. The revised version is fine.